# T-Cell Dynamics Predicts Prognosis of Patients with Hepatocellular Carcinoma Receiving Atezolizumab Plus Bevacizumab

**DOI:** 10.3390/ijms252010958

**Published:** 2024-10-11

**Authors:** Hye Won Lee, Suebin Park, Hye Jung Park, Kyung Joo Cho, Do Young Kim, Byungjin Hwang, Jun Yong Park

**Affiliations:** 1Department of Internal Medicine, Yonsei University College of Medicine, Seoul 03722, Republic of Korea; lorry-lee@yuhs.ac (H.W.L.); hjung31@yuhs.ac (H.J.P.); kyungjoo89@yuhs.ac (K.J.C.); drpjy@yuhs.ac (J.Y.P.); 2Insitute of Gastroenterology, Yonsei University College of Medicine, Seoul 03722, Republic of Korea; 3Yonsei Liver Center, Severance Hospital, Seoul 03722, Republic of Korea; 4Department of Clinical Drug Discovery & Development, Yonsei University College of Medicine, Seoul 03722, Republic of Korea; souvenir168@yonsei.ac.kr; 5Graduate School of Medical Science, Brain Korea 21 Project, Yonsei University College of Medicine, Seoul 03722, Republic of Korea; 6Department of Biomedical Sciences, Yonsei University College of Medicine, Seoul 03722, Republic of Korea

**Keywords:** atezolizumab, bevacizumab, hepatocellular carcinoma, immune dynamics, prognostic markers

## Abstract

Atezolizumab and bevacizumab show promise for treating hepatocellular carcinoma (HCC), but identifying responsive patients remains challenging, due to tumor heterogeneity. This study explores immune dynamics following this combination therapy. Between 2020 and 2023, 29 patients with advanced HCC who received atezolizumab plus bevacizumab at Severance Hospital, Seoul, were enrolled in this study. Peripheral blood mononuclear cells were analyzed using flow cytometry and statistical methods to assess immune alterations and identify biomarkers. Baseline characteristics showed a diverse HCC cohort with a mean age of 64 years and 82.8% male predominance. Absence of extrahepatic metastasis was associated with better overall survival. Immune responses revealed distinct CD4^+^ T-cell phenotypes between the ‘partial response (PR) + stable disease (SD)’ and ‘progressive disease (PD)’ groups, with an overall increase in CD8^+^ T-cell phenotypes. Patients with higher frequencies of CD8^+^PD-1^+^Ki-67^+^ T cells experienced significantly improved overall survival, while those with lower frequencies of CD4^+^Foxp3^+^PD-1^+^LAG3^+^ T cells also had notable survival benefits. These findings enhance the overall understanding of immune responses to this combination therapy, facilitating improved patient stratification and personalized therapeutic approaches for HCC.

## 1. Introduction

The combination of atezolizumab, an inhibitor of programmed death-ligand 1 (PD-L1), and bevacizumab, an anti-vascular endothelial growth factor (VEGF) agent, shows therapeutic promise for hepatocellular carcinoma (HCC) [1]. Although sorafenib and lenvatinib have long been used as first-line therapies [2], this innovative combination has shown considerable efficacy in clinical trials, leading to improved overall survival (OS) and progression-free survival (PFS). The synergistic effect of atezolizumab plus bevacizumab is based on their complementary mechanisms of action (i.e., targeting immune checkpoint inhibitors and angiogenesis, respectively) [3,4,5]. As HCC treatment evolves, an understanding of the effect of this combination therapy on immune dynamics will be more important for efforts to optimize patient outcomes.

Despite advances in immunotherapy for HCC, it is challenging to identify patients who are likely to respond favorably to atezolizumab plus bevacizumab [6,7]. Heterogeneity in the HCC patient population, at both molecular and immune levels, results in highly variable responses to treatment. Accurate prediction of treatment response is hindered by the lack of reliable biomarkers, as well as the intricate interplay between the tumor microenvironment and the host immune system. Solutions to these challenges would enable personalized therapeutic strategies and maximize the clinical benefit of atezolizumab plus bevacizumab in patients with HCC.

Investigation of the dynamics of immune cells after treatment with atezolizumab plus bevacizumab would provide insight into the mechanisms that underlie the treatment response and resistance in HCC. Few studies have focused on the modulation of T-cell subsets, including cytotoxic T cells and regulatory T cells, as well as myeloid-derived suppressor cells, in the context of this combination therapy [8]. Although progression to immunotherapy have been reported, no biomarkers can accurately predict outcomes after this treatment. Therefore, investigations of the dynamics of T-cell subsets would help to identify candidate biomarkers that predict the efficacy of atezolizumab plus bevacizumab.

In this study, we collected serial samples of peripheral blood mononuclear cells (PBMCs) from 29 patients with advanced HCC. We evaluated the effect of treatment with atezolizumab plus bevacizumab on the composition of immune cell populations, as well as OS.

## 2. Results

### 2.1. Baseline Characteristics

In total, 29 patients treated with atezolizumab and bevacizumab were analyzed (Table 1). The mean age of the patients was 63.8 ± 11.0 years, with a predominance of men (82.8%). Most patients (58.6%) had viral causes of HCC; 41.4% had non-B non-C (NBNC) HCC. Portal vein invasion and extrahepatic metastasis were observed in 41.4% and 55.2% of the patients, respectively. The median OS was 16.0 months (IQR 10.7–25.7) and 17 (58.6%) patients died due to liver-related events. The frequencies of maximal treatment responses were partial response (PR), 20.7%; stable disease (SD), 27.6%; and progressive disease (PD), 51.7%. Furthermore, 58.6% of the patients experienced a progression of HCC within 6 months. We compared the PR + SD group (*n* = 14) and the PD group (*n* = 15) in patients treated with atezolizumab plus bevacizumab (Table 2). The PR^+^SD group tended to be older (66.4 ± 6.4 vs. 61.4 ± 13.7 years, *p* = 0.230) and had a higher proportion of men (92.9% vs. 73.3%, *p* = 0.164) compared to the PD group. Laboratory findings showed no significant differences between the groups. The rate of progression within 6 months was significantly lower in the PR^+^SD group compared to the PD group (14.3% vs. 100%, *p* < 0.001), while the mortality rate did not differ significantly (64.3% vs. 53.3%, *p* = 0.550).

### 2.2. Effects of T-Cell Phenotypes on HCC Etiology, Extrahepatic Metastasis and Survival Outcomes

To investigate the relationship between baseline T-cell frequencies and the etiology of HCC, patients were categorized into groups based on viral and non-viral causes of HCC. Comparison of median baseline T-cell frequencies (Figure 1a) indicated significant differences in CD4^+^Foxp3^+^, CD4^+^Foxp3^−^, and CD8^+^LAG3^+^ between patients with viral and non-viral causes of HCC (Figure 1b). The frequency of CD4^+^Foxp3^−^ cells was higher in patients with viral HCC; the frequencies of CD4^+^Foxp3^+^ and CD8^+^LAG3^+^ cells were higher in patients with non-viral HCC. No other significant differences were noted, and no clear trends emerged.

Additionally, patients were divided into two groups based on the presence or absence of extrahepatic metastasis. Analysis revealed that patients without metastasis exhibited significantly higher frequencies of CD8^+^IFN-γ^+^ and CD8^+^TIM3^+^Ki-67^+^ T cells compared to those with metastasis (Figure 1c,d).

Furthermore, our investigation revealed significant correlations between each phenotype and cumulative survival rate, particularly for CD8^+^PD-1^+^Ki-67^+^ and CD4^+^Foxp3^+^PD-1^+^LAG3^+^ T cells (log-rank *p* = 0.011 and 0.044, respectively; Figure 1e,f). These results were obtained through a receiver operating characteristic analysis to determine optimal cutoff values, followed by stratifying the patients based on these cutoffs and performing a survival analysis. For CD8^+^PD-1^+^Ki-67^+^ T cells, patients were stratified into two groups using the optimal Ki-67^+^ frequency cutoff of 9.975%. Patients with a baseline frequency exceeding this cutoff showed improved OS, suggesting that a higher frequency of CD8^+^PD-1^+^Ki-67^+^ T cells at baseline is predictive of longer OS. For CD4^+^Foxp3^+^PD-1^+^LAG3^+^ T cells, the optimal LAG3^+^ frequency cutoff was 9.08%. Notably, patients with a baseline frequency below this cutoff exhibited improved OS. This suggests that a lower frequency of CD4^+^Foxp3^+^PD-1^+^LAG3^+^ T cells at baseline is predictive of longer OS.

### 2.3. Frequencies of CD4^+^ T-Cell Phenotypes before and after Treatment with Atezolizumab Plus Bevacizumab

We investigated the effect of atezolizumab plus bevacizumab on the frequencies of CD4^+^ T-cell phenotypes in 28 patients with HCC, comparing baseline and week 3 in 13 PR ^+^SD and 15 PD patients (Figure 2a). One patient within the PR^+^SD group did not have an available baseline sample; therefore, this patient was excluded from the CD4+ T cell phenotype analysis. Focusing on specific markers including Foxp3, PD-1, LAG-3, TIM-3, and Ki-67, we observed changes in the frequencies of diverse CD4+ T-cell phenotypes following treatment with atezolizumab plus bevacizumab.

The frequencies of CD4^+^ and CD4^+^Ki-67^+^ T cells increased in the PR^+^SD group but not in the PD group. The frequencies of CD4^+^PD-1^+^ and CD4^+^PD-1^+^Ki-67^+^ T cells increased in PR^+^SD and PD, but the increase in PD-1^+^Ki-67^+^ was statistically significant only in PR^+^SD (1.48% to 2.94%, *p* < 0.05; Figure 2b). Although the frequency of CD4^+^Foxp3^+^ regulatory T cells (Tregs) increased in PR^+^SD and PD, the frequencies of CD4^+^Foxp3^+^PD-1^+^ and CD4^+^Foxp3^+^PD-1^+^Ki-67^+^ T cells increased in PR^+^SD but decreased in PD. Conversely, the frequencies of CD4^+^Foxp3^−^PD-1^+^ and CD4^+^Foxp3^−^PD-1^+^Ki-67^+^ T cells increased in PR^+^SD and PD. CD4^+^Foxp3^+^PD-1^+^ and CD4^+^Foxp3^−^PD-1^+^ T cells showed significantly increased frequencies of Ki-67 expression in PR^+^SD (8.9% to 10.2%, *p* < 0.05; 12.9% to 20.2%, *p* < 0.01, respectively; Figure 2b), suggesting enhanced proliferation of these T-cell subsets.

We next focused on 11 treatment-naïve patients who received atezolizumab plus bevacizumab as the first-line treatment. (Appendix A). In this subset, consisting of 7 PR^+^SD and 4 PD, we observed similar trends in CD4^+^ T-cell phenotypes as compared to the entire cohort results; however, these trends were not statistically significant.

### 2.4. CD8^+^ T Cells with Inhibitory Receptors Are Reinvigorated by Atezolizumab Plus Bevacizumab

The frequencies of the majority of CD8^+^ T-cell phenotypes increased after treatment in both the PR^+^SD (*n* = 14) and PD (*n* = 15) groups (Figure 3a). In the PR^+^SD group, the frequencies of most CD8^+^ phenotypes increased after treatment; the exceptions were CD8^+^, CD8^+^LAG3^+^, and CD8^+^PD-1^+^LAG3^+^ T cells. In the PD group, all CD8^+^ T-cell phenotypes increased following treatment. To illustrate subtle distinctions between PR^+^SD and PD, we represented changes in median cell frequencies from baseline to week 3 as fold changes (fold), derived from the ratio of week 3-to-baseline frequencies. We identified significant changes in CD8^+^ T cells with inhibitory receptor and their proliferation status (Figure 3b). In the PR^+^SD group, the frequency of PD-1 expression in CD8^+^T cells significantly increased (10.85% to 15.95%, 1.37-fold; *p* < 0.05). Additionally, the frequencies of Ki-67^+^ in CD8^+^LAG3^+^ cells (0.37% to 1.03%, 2.80-fold; *p* < 0.01), CD8^+^PD-1^+^ cells (8.48% to 31.25%, 3.69-fold; *p* < 0.01), and CD8^+^TIM3^+^ cells (0.35% to 0.52%, 1.47-fold; *p* < 0.05) increased. In the PD group, the frequency of PD-1 expression in CD8^+^ cells (15.3% to 21%, 1.47-fold; *p* < 0.05) and the frequencies of Ki-67 in CD8^+^LAG3^+^ (0.42% to 0.86%, 2.05-fold, *p* < 0.05), CD8^+^PD-1^+^ (8.73% to 17.7%, 2.03-fold; *p* < 0.01), and CD8^+^TIM3^+^ (0.37% to 0.57%, 1.54-fold; *p* < 0.05) cells increased. Notably, the fold change in the frequency of Ki-67 expression among CD8^+^LAG3^+^ and CD8^+^PD-1^+^ T cells was greater in the PR^+^SD group than in the PD group.

Analysis of treatment-naïve patients yielded similar results (Appendix A). There were significant post-treatment increases in the frequencies of PD-1 expression in CD8^+^ cells and Ki-67 expression in CD8^+^, CD8^+^PD-1^+^, CD8^+^LAG3^+^, and CD8^+^TIM3^+^ cells. These results show that treatment with atezolizumab plus bevacizumab reinvigorates CD8^+^ T cells, particularly those expressing inhibitory receptors such as PD-1, LAG3, and TIM3, along with Ki-67, consistent with the observations in the entire cohort. Additionally, there were significant differences in the median baseline frequencies of CD8^+^ T and PD-1 expression in CD8^+^ T cells between PR^+^SD (24.5% and 10.3%, respectively) and PD (13.9% and 14.75%, respectively) among treatment-naïve patients (Appendix A). This result indicates the potential role of these cells as baseline biomarkers for first-line treatment responses. Larger cohort studies are needed to confirm these associations.

## 3. Discussion

The lack of reliable biomarkers for predicting the response to atezolizumab-plus-bevacizumab therapy is a challenge in clinical practice. Despite advances in precision medicine, efforts to identify patients who will benefit from immunotherapy are problematic because of the heterogeneity of HCC and the complexity of its immunobiology. Several candidate biomarkers—including PD-L1 expression, tumor mutational burden, and immune cell infiltration—have been investigated in the context of immunotherapy for HCC [9]. However, the findings are inconsistent, and few validation studies have been performed. Furthermore, the dynamic interplay among the tumor microenvironment, host immune response, and treatment modalities hinders biomarker discovery and validation. Collaborative efforts involving multi-omics approaches, robust clinical trials, and large-scale validation studies are needed to identify and validate predictive biomarkers that can guide treatment decision-making and improve the outcomes of patients with HCC undergoing atezolizumab-plus-bevacizumab therapy.

In our exploration of immune dynamics, we identified a marked increase in the proliferation of PD-1-expressing CD4^+^ T-cell subsets. These findings align with a previous report indicating a significant increase in the proportion of Ki-67^+^ cells among both Foxp3^−^ and Foxp3^+^CD4^+^ T cells in response to PD-1 blockade, peaking at week 3, particularly within the PD-1^+^ subset of each population in solid tumors [10]. This supports our results and emphasizes the dynamic CD4^+^ T-cell response to atezolizumab-plus-bevacizumab treatment. Previous studies reported that PD-1^+^ and Foxp3^+^ Tregs have important roles in the immune dysfunction exhibited by patients with HCC [11,12]. Elevated levels of circulating Tregs have been linked to increased mortality and reduced survival duration in patients with HCC [13]. Additionally, targeted depletion of highly suppressive Tregs may help to restore the function of T-effector cells in patients with advanced HCC [12]. A decrease in the number of CD4^+^Foxp3^+^PD-1^+^ T cells corresponds to an improved survival rate in patients with HCC who receive sorafenib [14]. Similarly, decreased LAG3 expression on CD4^+^Foxp3^+^ T cells was associated with significantly longer PFS in patients with multiple myeloma [15]. In the present study, a lower baseline frequency of CD4^+^Foxp3^+^PD-1^+^LAG3^+^ T cells was associated with improved OS. These findings suggest that analogous immunological markers have potential for predicting the outcomes of patients with various malignancies.

The frequency of Ki-67^+^ cells among exhausted CD8^+^ T cells were substantially enhanced in the PR^+^SD group. There were increases in the frequencies of some CD8^+^ T-cell phenotypes after treatment in the entire cohort (*n* = 29) and the treatment-naïve subset, indicating reinvigoration of these cells; the reinvigoration was greatest among cells expressing inhibitory receptors (PD-1, LAG3 and TIM3) and Ki-67. Among these, the increase in the frequency of Ki-67 expression was greatest in CD8^+^LAG3^+^ and CD8^+^PD-1^+^ T cells in the PR^+^SD group after treatment. This suggests that CD8^+^LAG3^+^Ki-67^+^ and CD8^+^PD-1^+^Ki-67^+^ hold promise as potential biomarkers for distinguishing PR^+^SD from PD; however, further investigation into their predictive role is needed. In patients with renal cell carcinoma who received atezolizumab plus bevacizumab, the frequency of intratumoral CD8^+^ T cells was increased as a result of augmented trafficking and infiltration of proliferating CD8^+^ T cells [16]. In patients with melanoma, exhausted CD8^+^ T-cell phenotypes were reinvigorated by anti-PD-1 treatment [10]. In the context of anti-PD-1 therapy in cancer treatment, the ratio of Ki-67^+^ cells among PD-1^+^CD8^+^ T cells after treatment compared with baseline is predictive of long-term treatment outcomes [17]. Our study further expanded upon this understanding by observing alterations in Ki-67 expression not only within PD-1^+^ subsets, but also among TIM3^+^ or LAG3^+^ subsets. Recent studies have examined the predictive abilities of PD-L1 expression and tumor-infiltrating lymphocytes in HCC immunotherapy. Our study of the proliferation marker Ki-67 sheds light on immune cell activity and functionality in the tumor microenvironment. Exploring CD8^+^PD-1^+^Ki-67^+^ T cells provides insight into the dynamic interplay between the immune response and survival outcomes. Additionally, our investigation of CD8^+^PD-1^+^Ki-67^+^ as a potential biomarker contributes to the emerging exploration of immune checkpoint markers in predicting patient survival outcomes after immunotherapy.

Here, we investigated the therapeutic potential of atezolizumab plus bevacizumab for HCC; we sought to address challenges associated with the heterogeneity of the HCC patient population. A high frequency of CD8^+^PD-1^+^Ki-67^+^ T cells was associated with longer OS. Indeed, it has been reported that an early increase in the frequency of Ki-67^+^ cells among CD8^+^PD-1^+^ T cells after anti-PD-1 treatment is associated with better clinical outcomes in patients with solid tumors [18]. A low frequency of CD4^+^Foxp3^+^PD-1^+^LAG3^+^ T cells was also linked to a significant survival benefit. Our findings provide insight into early immune alterations that occur after treatment with atezolizumab plus bevacizumab, facilitating improved patient stratification and the development of personalized therapies for HCC.

Non-viral etiologies of HCC (e.g., nonalcoholic fatty liver disease [NAFLD]) have clinical characteristics, molecular and immune profiles, and responses to treatment that are distinct from those of viral HCC [19]. Combined metabolic abnormalities can create an immunosuppressive tumor microenvironment, characterized by increased levels of pro-inflammatory cytokines, recruitment of myeloid-derived suppressor cells, and impaired function in cytotoxic T cells. In the present study, non-viral HCC influenced the outcomes of treatment with atezolizumab plus bevacizumab. Correlation analysis indicated a trend towards poorer survival outcomes in non-viral HCC cases. Additionally, significantly higher baseline frequencies of CD4^+^Foxp3^+^ and CD8^+^LAG3^+^ cells were observed in patients with non-viral HCC. These findings highlight the heterogeneity of HCC and the need for treatment approaches that are customized according to tumor biology and underlying liver disease etiology. Future research should aim to validate these findings in larger cohorts and elucidate the mechanisms underlying the association between non-viral HCC and response to treatment with atezolizumab plus bevacizumab.

Our study sheds light on the relationship between T-cell phenotype and HCC extrahepatic metastasis. We found that HCC patients without extrahepatic metastasis exhibited higher frequencies of CD8^+^IFN-γ^+^ and CD8^+^TIM3^+^Ki-67^+^ T cells. This suggests a more complex immune profile, as CD8^+^IFN-γ^+^ T cells indicate activation, while CD8^+^TIM3^+^Ki-67^+^ T cells reflect an exhausted status. Further research is needed to fully understand the roles of these T-cell subsets in extrahepatic metastasis. Additionally, circulating tumor DNA (ctDNA) has potential as a biomarker for response to immunotherapy. A recent study investigated the dynamics of ctDNA in patients with HCC [20]. The persistence of mutations during the initial treatment was associated with a higher rate of progression compared to those whose mutations was disappeared. Although the heterogeneity of HCC and the low incidence of molecular alterations need to be addressed, several studies suggest the potential of ctDNA as a biomarker [21,22].

This study had several limitations. The small number of patients, a common issue in studies of advanced HCC treated with immunotherapy, requires a cautious interpretation of the results. Although we collected serial blood samples after immunotherapy, the small sample size is a limitation of this study. Additionally, our focus on early immunologic changes, which enabled the identification of early responders, may not have fully captured the immunologic changes that occur during immunotherapy. These limitations highlight the need for larger, multicenter studies with longer follow-up periods to validate our findings. Finally, the absence of liver biopsy samples in our study constituted a limitation. Liver biopsy is important for assessing the response to immunotherapy in patients with HCC; thus, there is a need for access to tissue samples. Our use of serum samples from HCC patients is consistent with the trend towards biomarkers exhibiting greater accessibility that can be evaluated in a minimally invasive manner. Future research should explore the feasibility of alternative sampling methods, such as liquid biopsy, for investigations of immunologic changes in HCC during immunotherapy.

In conclusion, our study provides insight into early immune alterations after treatment with atezolizumab plus bevacizumab. Additionally, these findings enhance the overall understanding of immune responses to this combination therapy, facilitating improved patient stratification and personalized therapeutic approaches for HCC.

## 4. Materials and Methods

### 4.1. Study Population

Between 2020 and 2023, we enrolled patients with advanced HCC who had been treated with atezolizumab plus bevacizumab at Severance Hospital, Seoul, Republic of Korea. The diagnosis of HCC was made in accordance with current guidelines [23], and serial serum samples were collected from the patients. The inclusion criteria were preserved liver function (Child–Pugh score A) and good performance status. Selection of atezolizumab-plus-bevacizumab treatment was based on the judgment of the attending physician, informed by clinical practice guidelines and with input from a multidisciplinary team. The exclusion criteria were age ≤ 18 years, a history of malignancy other than HCC, and a history of liver transplantation.

Patients received 1200 mg of atezolizumab plus 15 mg/kg body weight of bevacizumab intravenously every 3 weeks. Treatment response was evaluated by analyzing the levels of tumor markers and by performing computed tomography or magnetic resonance imaging every 3 months. Tumor response was evaluated using the modified RECIST criteria, which assess treatment response by measuring the longest diameters of all arterial-enhancing lesions [24]. PR was defined as a reduction of more than 30% in the viable portion of the tumor, while PD was defined as an increase of more than 20% in the viable portion of the tumor or the development of new lesions. Safety was continuously evaluated according to the National Cancer Institute Common Terminology Criteria for Adverse Events, ver. 4.0. The study was approved by the Independent Institutional Review Board of Severance Hospital (Number: 4-2023-1591) and conformed to the ethical guidelines of the 1975 Declaration of Helsinki. We obtained written informed consent from all participants in this study.

### 4.2. Sample Collection and Processing

Blood sampling was carried out before and 3 weeks after atezolizumab-plus-bevacizumab treatment. To isolate PBMCs, peripheral blood samples were collected in anticoagulant-coated tubes and carefully layered atop a Ficoll–Paque density gradient solution (Cytiva, Marlborough, MA, USA). After centrifugation, the mononuclear cell layer was collected, washed twice, resuspended in PBS, resuspended in freezing medium, and stored at −70 °C, then placed in liquid nitrogen until analysis.

### 4.3. Flow Cytometry

Cells washed in PBS were incubated with Live/Dead Cell Viability Dye (Thermo Fisher Scientific, Bedford, MA, USA). Multicolor flow cytometry was performed using commercially available antibodies against CD3 (UCHT1), CD4 (OKT4), TIM3 (F38-2E2), and Foxp3 (PCH101) from Thermo Fisher Scientific; CD8 (RPA-T8) and IFN-γ (B27) from BD Biosciences; and PD-1 (EH12.2H7), LAG3 (7H2C65), and Ki-67 from BioLegend. After cells had been stained for cell-surface markers, they were washed twice with FACS buffer and incubated in fixation/permeabilization buffer (Invitrogen, Waltham, MA, USA) at 4 °C for 30 min. After two washes in permeabilization wash buffer, cells underwent intracellular staining for Foxp3, IFN-γ, and/or Ki-67 for 30 min at 4 °C. Cells were washed again, then subjected to flow cytometry using an SA3800 (Sony Biotechnology Inc., San Jose, CA, USA) with FlowJo software ver. 10.6.2 (Tree Star Inc., Ashland, OR, USA), according to a predefined gating strategy. The reagents used for flow cytometry are listed in Appendix A. All data generated or analyzed during this study are included in this article. Further enquiries can be directed to the corresponding author.

### 4.4. Statistical Analysis

Continuous data are presented as means ± standard deviations or medians (interquartile ranges); categorical data are presented as counts and percentages. For continuous data, Student’s *t*-test was used to identify differences between groups. When the assumption of normality was violated, a nonparametric test (Mann–Whitney or Wilcoxon signed-rank test) was performed. Differences between categorical variables were examined by the chi-squared test or Fisher’s exact test, as appropriate. Two-sided *p*-values < 0.05 were considered statistically significant. Cox proportional hazard models were used to estimate risk factors for OS. Statistical analysis was performed using SPSS software (ver. 25) and R software (ver. 4.3.1).

## Figures and Tables

**Figure 1 ijms-25-10958-f001:**
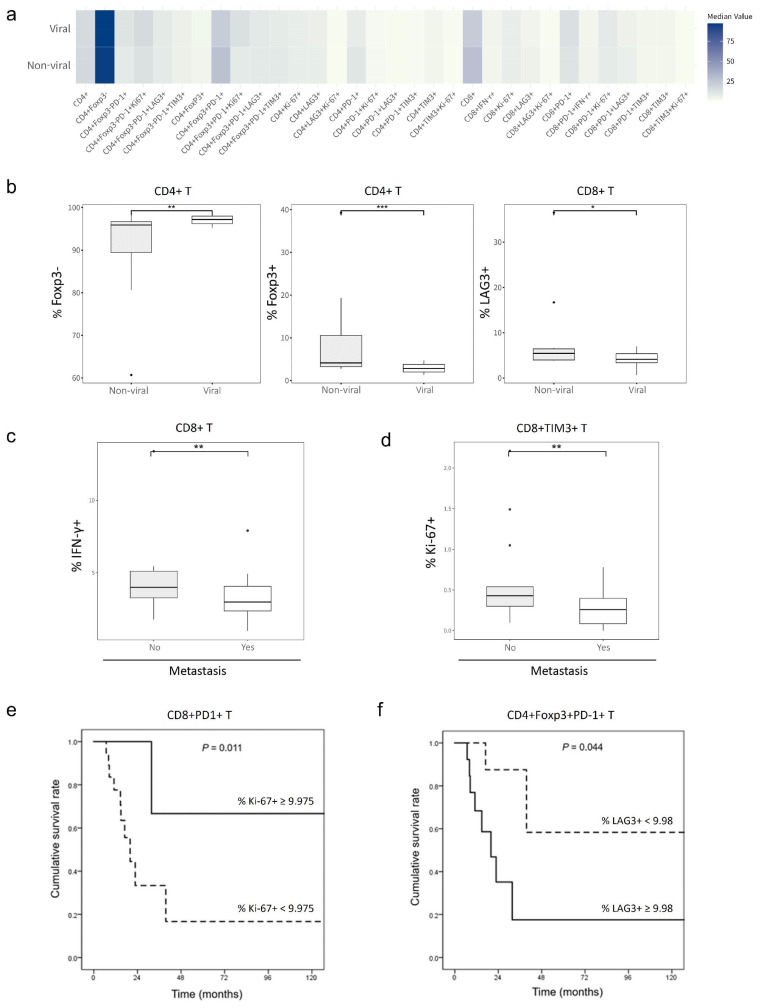
Frequencies of T-cell phenotypes at baseline according to HCC etiology and survival. (**a**) Heatmap of the median baseline frequencies of 32 T-cell phenotypes according to HCC etiology (viral vs. non-viral). (**b**) Box plots of the baseline frequencies of CD4^+^Foxp3^−^, CD4^+^Foxp3^+^, and CD8^+^LAG3^+^ T cells according to HCC etiology (viral, *n* = 17 vs. non-viral, *n* = 12). (**c**) Box plot of the baseline frequencies of CD8^+^IFN-γ T cells. (**d**) Box plot of the baseline frequencies of CD8^+^TIM3^+^Ki-67^+^ T cells. (**e**) Cumulative survival curve based on the frequency of Ki-67^+^ T cells within CD8^+^PD-1^+^ T-cell population. (**f**) Cumulative survival curve based on the frequency of LAG3^+^ T cells within CD4^+^Foxp3^+^PD-1^+^ T-cell population. *** *p* < 0.001,** *p* < 0.05, * *p* < 0.1 according to Mann–Whitney U-test.

**Figure 2 ijms-25-10958-f002:**
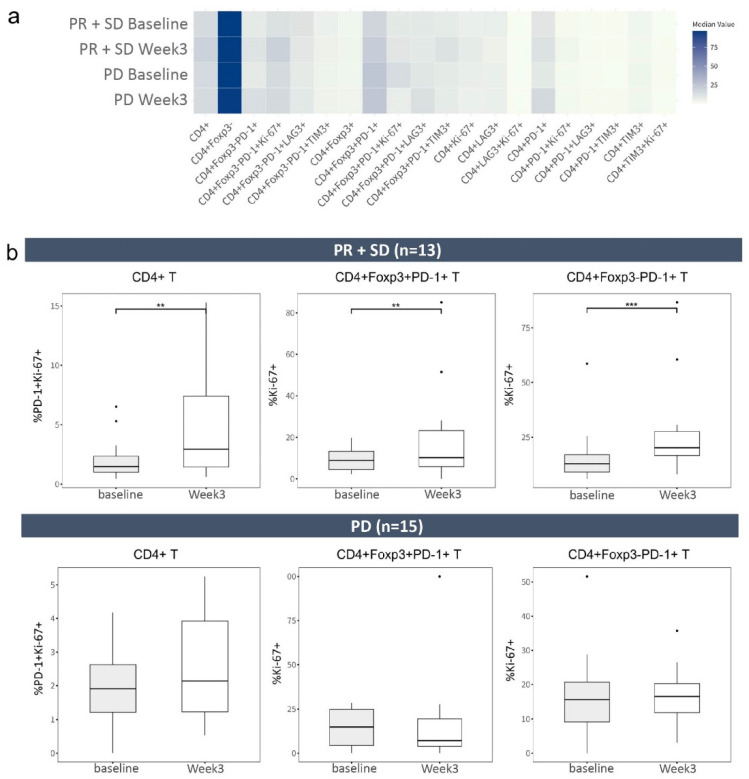
Effect of treatment with atezolizumab and bevacizumab on the frequencies of CD4^+^ T-cell phenotypes. (**a**) Heatmap of the median frequencies of 20 CD4^+^ T-cell phenotypes in PR^+^SD and PD at baseline and week 3 in a cohort of 28 patients who underwent treatment with atezolizumab plus bevacizumab. Color intensities indicate median values, ranging from light (low) to dark (high). (**b**) Box plots of the frequencies of CD4^+^PD-1^+^Ki-67^+^, CD4^+^Foxp3^+^PD-1^+^Ki-67^+^, and CD4^+^Foxp3-PD-1^+^Ki-67^+^CD4^+^ T cells in PR^+^SD (*n* = 13) and PD (*n* = 15) at baseline and week 3. Patient 8,490,836 within the PR^+^SD group did not have an available baseline sample; therefore, this patient was excluded from the analysis. PR, partial response; SD, stable disease; PD, progressive disease. *** *p* < 0.001, ** *p* < 0.05 according to Wilcoxon matched-pair signed-rank test.

**Figure 3 ijms-25-10958-f003:**
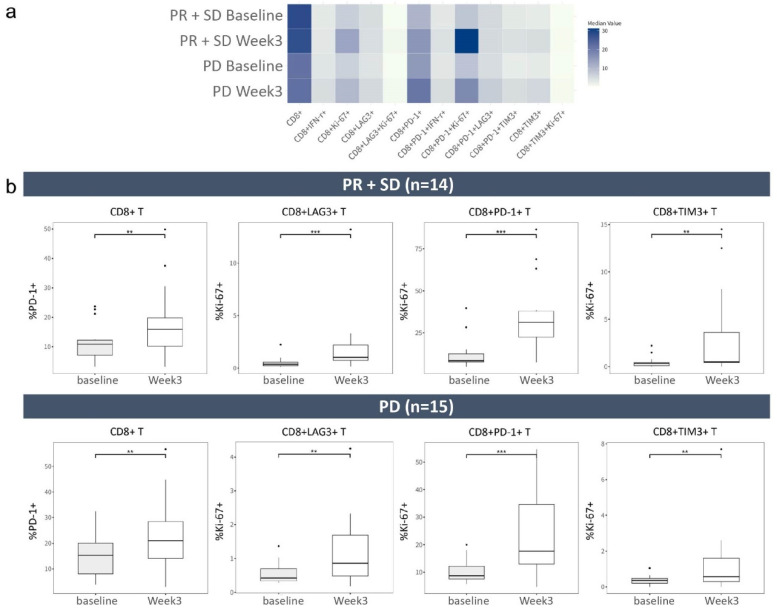
Increase in the majority of CD8^+^ T-cell phenotypes after treatment. (**a**) Heatmap of the median frequencies of 12 CD8^+^ T-cell phenotypes in PR^+^SD and PD groups at baseline and week 3 in a cohort of 29 patients. Color intensities indicate median values, ranging from light (low) to dark (high). (**b**) Box plots of the frequencies of CD8^+^PD-1^+^, CD8^+^LAG^+^Ki-67^+^, CD8^+^PD-1^+^Ki-67^+^, and CD8^+^TIM3^+^Ki-67^+^ T cells in PR^+^SD (*n* = 14) and PD (*n* = 15) at baseline and week 3. *** *p* < 0.001, ** *p* < 0.05 according to Wilcoxon matched-pair signed-rank test.

**Table 1 ijms-25-10958-t001:** Baseline characteristics.

Characteristic	Total (*n* = 29)
Age, years	63.8 ± 11.0
Male sex	24 (82.8)
AST, IU/L	54.0 (36.0–76.5)
ALT, IU/L	22.0 (16.5–47.0)
Total bilirubin, mg/dL	0.8 (0.7–1.2)
Platelet, 10^3^/μL	156 (100–205)
Albumin, g/dL	4.1 (3.8–4.4)
PT INR	1.03 (0.99–1.10)
AFP, ng/mL	243.8 (13.2–6855.6)
PIVKA-II, mAU/mL	2047 (136–39,774)
Causes of HCC	
Viral	17 (58.6)
NBNC	12 (41.4)
Portal vein invasion	12 (41.4)
Extrahepatic metastasis	16 (55.2)
Presence of varix	8 (27.6)
Best response	
PR	6 (20.7)
SD	8 (27.6)
PD	15 (51.7)
Progression within 6 months	17 (58.6)

Values are presented as *n* (%), mean ± standard deviation, or median (IQR). Abbreviations: AST, aspartate aminotransferase; ALT, alanine aminotransferase; PT INR, prothrombin-time international normalized ratio; AFP, α-fetoprotein; PIVKA-II, protein-induced vitamin K antagonist-II; HCC, hepatocellular carcinoma; NBNC, non-B non-C; PR, partial response; SD, stable disease; PD, progressive disease.

**Table 2 ijms-25-10958-t002:** Comparison of PR^+^SD and PD after treatment with atezolizumab plus bevacizumab.

	PR^+^SD (*n* = 14)	PD (*n* = 15)	*p*-Value
Age, years	66.4 ± 6.4	61.4 ± 13.7	0.230
Male sex	13 (92.9)	11 (73.3)	0.164
AST, IU/L	49.5 (31.5–93.5)	60.0 (36.0–72.0)	0.727
ALT, IU/L	28.5 (10.8–53.5)	19.0 (17.0–32.0)	0.631
Total bilirubin, mg/dL	0.9 (0.6–1.3)	0.8 (0.7–1.1)	0.982
Platelet, 10^3^/μL	159 (113–245)	134 (90–182)	0.275
Albumin, g/dL	4.0 (3.8–4.3)	4.3 (3.6–4.5)	0.357
PT INR	1.02 (0.99–1.08)	1.07 (0.98–1.12)	0.358
AFP, ng/mL	23.9 (6.2–3483.1)	1162.7 (59.8–34,229.5)	0.070
AFP ≥ 200 ng/mL	4 (28.6)	7 (46.7)	0.316
PIVKA-II, mAU/mL	1099 (67–75,000)	2237 (327–11,661)	0.827
Causes of HCC			0.096
Viral	6 (42.9)	11 (73.3)	
NBNC	8 (57.1)	4 (26.7)	
Portal vein invasion	6 (42.9)	6 (40.0)	0.876
Extrahepatic metastasis	9 (64.3)	7 (46.7)	0.340
Presence of varix	5 (35.7)	3 (20.0)	0.344
Progression within 6 months	2 (14.3)	15 (100)	<0.001
Death	9 (64.3)	8 (53.3)	0.550

Values are presented as *n* (%), mean ± standard deviation, or median (IQR). Abbreviations: AST, aspartate aminotransferase; ALT, alanine aminotransferase; PT INR, prothrombin-time international normalized ratio; AFP, α-fetoprotein; PIVKA-II, protein-induced vitamin K antagonist-II; HCC, hepatocellular carcinoma; NBNC, non-B non-C; PR, partial response; SD, stable disease; PD, progressive disease.

## Data Availability

The original contributions presented in the study are included in the article/Appendix A; further inquiries can be directed to the corresponding authors.

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
