# Peer review of "T-Cell Dynamics Predicts Prognosis of Patients with Hepatocellular Carcinoma Receiving Atezolizumab Plus Bevacizumab"

_ijms, 2024, doi:10.3390/ijms252010958_

Round 1
Reviewer 1 Report
Comments and Suggestions for Authors
The authors conduct an impressive study looking at peripheral immune alterations in patients who responded to versus did not respond to IO for HCC.
I have the following suggestions
1. The terminology and abbreviations used are confusing. If you call something a partial versus non-responder, the abbreviations should be (PR vs NR). However, you say non responder, but then say PD which I think means progressive disease. I know this seems minor but it makes the paper extremely hard to comprehend.
2. ABSTRACT Stating that the progression rate was higher in nonresponders is a flawed analysis. Of course it is - that is how you defined progression. Remove or adjust this.
3. DISCUSSION same issue: "Furthermore, the rate of progression within 6 months was significantly lower in responders (defined as PR and SD) compared to non-responders." The discussion should not be whether responding to IO improves outcomes. This is well shown. The point should be that responders have XYZ immune alterations and that we can both target these and use them to screen patients for who should get IO.
3. I recommend discussion of other research assessing biomarkers for response to IO. To my knowledge this is primarily ctDNA. I am curious the authors thoughts on ctDNA and blood-based tumor mutational burden in this sense. Did the authors look at this? There have been studies describing response to IO and IO+LRT measured by ctDNA that might be useful
5. I recommend the authors follow these patients and do serial testing (1-week, 2-week, 3-week, 90-days etc). I am curious both the utility of these signatures in surveillance and also the dynamics of change in the weeks after IO. THis is a suggestion for a future study not required for this publication.
Comments on the Quality of English LanguageThe abbreviations chosen are difficult otherwise okay
Author Response
Response to reviewers’ comments
We are profoundly appreciative that our manuscript ijms-3244021, titled “T cell Dynamics Predicts Prognosis of Patients with Hepatocellular Carcinoma Receiving Atezolizumab Plus Bevacizumab”, has been given the opportunity for revision for publication in International Journal of Molecular Sciences. We have carefully considered the valuable comments and constructive feedback provided by the referees and the editor and have made great efforts to improve the manuscript accordingly. The following are point-by-point answers to specific questions raised by the reviewer. We hope that the revised version of the manuscript meets the priority required for publication.
Reviewer: 1
The authors conduct an impressive study looking at peripheral immune alterations in patients who responded to versus did not respond to IO for HCC.
I have the following suggestions
- The terminology and abbreviations used are confusing. If you call something a partial versus non-responder, the abbreviations should be (PR vs NR). However, you say non responder, but then say PD which I think means progressive disease. I know this seems minor but it makes the paper extremely hard to comprehend.
Response) Thank you for your valuable comments. To avoid confusion, we have revised the term ‘responder’ to ‘partial response and stable disease (PR+SD)’ and ‘non-responder’ to ‘progressive disease (PD)’. These changes have been applied consistently throughout the manuscript.
- ABSTRACT Stating that the progression rate was higher in nonresponders is a flawed analysis. Of course it is - that is how you defined progression. Remove or adjust this.
Response) Tumor response was evaluated using the modified RECIST criteria, which assess treatment response by measuring the longest diameters of all arterial enhancing lesions (reference: Modified RECIST (mRECIST) assessment for hepatocellular carcinoma. Semin. Liver Dis. 2010; 30: 52–60). Partial response (PR) was defined as a reduction of more than 30% in the viable portion of the tumor, while progressive disease (PD) was defined as an increase of more than 20% in the viable portion of the tumor or the development of new lesions. We have added this information to the Methods section and removed the corresponding sentence from the Abstract.
- DISCUSSION same issue: "Furthermore, the rate of progression within 6 months was significantly lower in responders (defined as PR and SD) compared to non-responders." The discussion should not be whether responding to IO improves outcomes. This is well shown. The point should be that responders have XYZ immune alterations and that we can both target these and use them to screen patients for who should get IO.
Response) We have removed the following sentence from the Discussion. As you suggested, the focus of our study is on highlighting that immune alterations may be useful in selecting patients who are likely to respond well to immunotherapy. However, further studies with larger patient cohorts are need to support this point. This has been addressed in the Discussion section (page 15, lines 281-289).
- I recommend discussion of other research assessing biomarkers for response to IO. To my knowledge this is primarily ctDNA. I am curious the authors thoughts on ctDNA and blood-based tumor mutational burden in this sense. Did the authors look at this? There have been studies describing response to IO and IO+LRT measured by ctDNA that might be useful
Response) ctDNA has the potential to be used as a biomarker for response to immunotherapy. A recent study investigated the dynamics of ctDNA in patients with HCC (reference: Circulating tumour DNA in patients with hepatocellular carcinoma across tumour stages and treatments; Gut. 2024 Aug). The baseline cfDNA concentration or presence of mutations in ctDNA did not correlated with overall survival. However, the persistence of mutations during the initial treatment was associated with a higher rate of progression compared those whose mutations was disappeared. Although the heterogeneity of HCC and the low incidence of molecular alterations need to be addressed, several studies suggest the potential of ctDNA as a biomarker (reference: Circulating tumor DNA (ctDNA) as a biomarker of response to therapy in advanced Hepatocellular carcinoma treated with Nivolumab. Cancer Biomarkers 41; Circulating Cell-Free DNA Profiling Predicts the Therapeutic Outcome in Advanced Hepatocellular Carcinoma Patients Treated with Combination Immunotherapy. Cancers 2022 Jul). We have added this information in the Discussion section (page 16, lines 306-312).
- I recommend the authors follow these patients and do serial testing (1-week, 2-week, 3-week, 90-days etc). I am curious both the utility of these signatures in surveillance and also the dynamics of change in the weeks after IO. THis is a suggestion for a future study not required for this publication.
Response) Thank you for your comment. We will plan to conduct further studies using serial sample.
Reviewer 2 Report
Comments and Suggestions for Authors
The study was well written and properly conducted. My main concern is on the very limited sample size that does not allow to draw strong and reliable conclusions on the topic.
Moreover, how were the patients matched between the two groups?
The authors should provide more insights on the current state of the art of the systemic therapies for HCC, for example citing the recent SRMA: PMID: 34017396
Author Response
Response to reviewers’ comments
We are profoundly appreciative that our manuscript ijms-3244021, titled “T cell Dynamics Predicts Prognosis of Patients with Hepatocellular Carcinoma Receiving Atezolizumab Plus Bevacizumab”, has been given the opportunity for revision for publication in International Journal of Molecular Sciences. We have carefully considered the valuable comments and constructive feedback provided by the referees and the editor and have made great efforts to improve the manuscript accordingly. The following are point-by-point answers to specific questions raised by the reviewer. We hope that the revised version of the manuscript meets the priority required for publication.
Reviewer: 2
The study was well written and properly conducted. My main concern is on the very limited sample size that does not allow to draw strong and reliable conclusions on the topic.
Moreover, how were the patients matched between the two groups?
The authors should provide more insights on the current state of the art of the systemic therapies for HCC, for example citing the recent SRMA: PMID: 34017396
Response) Thank you for your valuable comments. We acknowledge the limitation of the small sample size. However, we were able to observe the immune alterations between the PR+SD and PD groups. We hope that this pilot data will provide a foundation for future studies exploring its use as a biomarker. As shown in Table 2, the two groups exhibited similar baseline characteristics, and we did not perform matching for these groups in this study. Additionally, we have included an excellent article (PMID: 34017396) to help readers the current state of systemic therapy for HCC.
Round 2
Reviewer 1 Report
Comments and Suggestions for Authors
the authors have made changes as requested and I recommend publication.
Reviewer 2 Report
Comments and Suggestions for Authors
The revised manuscript is OK. Thank you!